# Impact of Implementing a Deodorization System on the Functioning of a Waste Management Plant

**Łukasz Szałata**

Department of Environmental Protection Engineering, Wroclaw University of Science and Technology, 50-377 Wrocław, Poland; lukasz.szalata@pwr.edu.pl; Tel.: +48-71-320-2500

**Abstract:** This article is an analysis of the impact of implementing technical and technological solutions in the form of a deodorization system, which was carried out on the basis of a case study in the years 2015 and 2019 of an existing waste management plant named located in the region of Lower Silesia. The method for determining the impact was based on an analytical method for estimating changes in plant operation parameters in relation to the emission of odor compounds, determined at selected reference points, during a period of normal operation before and directly after the implementation of the deodorization system. The purpose of this work is the evaluation of the impact of the deodorization system on the parameters determining the functioning of the plant—a mechanical-biological waste treatment facility in order to determine sustainability factors and their relative changes.

**Keywords:** sustainability development; waste management plant; deodorization system; environmental effects; odor emission

---

## 1. Introduction

Many business entities characterized by odor nuisance resulting from technological processes currently perform a range of organizational and technological activities in order to meet environmental standards, including limiting the impact of atmospheric emissions from, among other things, odor-active compounds [1]. A business operation that involves the emission of odorous compounds is often legally obliged to implement technical solutions in order to reduce emissions [2]. This obligation often incurs additional financial expenses for the plant, which sometimes raises doubts as to the economic justification for implementing such technical and technological solutions. Such doubts are mainly caused by a lack of awareness on the part of the plant operator as regards possible technical solutions. Technical solutions that modify the existing technological system may cause changes in individual processes, thus creating not only negative but also positive effects [3]. The study is a relatively new approach in the context of determining the potential impact of deodorization systems, as an element limiting the impact on the environment. The derived results can be a reference value in determining the aspects of sustainable development that will allow for precise building of local space. Moreover, it can be the basis for selecting optimal technical solutions for this type of industry. The decision of which technology to select is made more complex, because technology performance is specified by a large number of parameters for which there are, as yet, no industry-wide standards [4]. Even for managers with high technical competency, the number of available technologies and their wide range of performance and cost is overwhelming [5]. The assumption of the deodorization system is a full correlation between the aspects of environmental management, the economic and environmental effectiveness of the implemented solutions, as well as justified environmental factors for the environment of a given economic entity, in this case in terms of reducing the odor impact [6,7].

Odors are often of a natural nature and are not clearly defined as toxic or dangerous to humans; however, their intensity may cause clear psychological discomfort in people. However, despite not being a direct cause of disease, long-term exposure to such odorants can have a negative effect on human well-being, causing nausea, headaches, insomnia, loss of appetite, respiratory problems, and irrational behavior [8]. Thus, treatment of gaseous emissions is an important measure to protect both public health and the environment [9]. The issue of odors emission is an important issue in terms of the potential environmental impact of plants of various technology industries. The goals of sustainable development are the priority directions of human development [10].

## 2. Materials and Methods

### 2.1. Analytical Method

The analytical method consists in determining the variability of the values of parameters that dictate the efficiency of the plant's operation (Table 1), based on measurements points (Table 2) and also such as plant performance and demand for electricity, in relation to the variability of odor compound emission levels (Table 3). The variability of these levels was calculated on the basis of changes in the concentration of these compounds over a certain period of the plant's operation, i.e., before and after the implementation of the deodorization system [11]. To determine the extent of the impact and the measurable total emission reduction taking into account spatial conditions, airborne pollutant spread was modelled using Operat FB software. The model used, which is consistent with the reference methodology described in the Regulation of the Minister of the Environment of 26 January 2010 on reference values for certain substances in the air, is based on calculations for the Gaussian "plume" model, shaped by wind and diffusion processes.

**Table 1.** Process parameters [source: data obtained directly from the operator].

| Process Parameter | | Before Implementation of the Deodorization System (Period I) | After Implementation of the Deodorization System (Period II) |
|---|---|---|---|
| Actual plant performance | Process D8 | 59,696.1 Mg/year | 74,616.74 Mg/year |
| | Process R3 | 3914.60 Mg/year | 4989.70 Mg/year |
| Energy consumption | | 488,816 kWh | 470,198 kWh |

**Table 2.** Measuring points.

| Emitter Name | Point No. | Location |
|---|---|---|
| EP-1 | 1 | Green waste storage yard |
| EP-2 | 2 | Compost storage yard |
| EP-3 | 3 | Mixed waste storage yard |
| E-4 | 4 | Sorting facility |

**Table 3.** Results of measurements of atmospheric concentrations of substances within the plant premises in periods I, II.

| Point No. | | 3 | 2 | 4 | 1 |
|---|---|---|---|---|---|
| Emitter Name | | EP1 | EP2 | EP3 | E4 |
| | *Before the Implementation of the Deodorization System (Period I)* | | | | |
| | Ammonia | 0.210 | 0.085 | 0.400 | 0.110 |
| | $H_2S$ | <0.028 | <0.028 | <0.028 | <0.028 |
| | Ethyl mercaptan | 0.160 | 0.055 | 0.055 | <0.006 |
| | Butyl mercaptan | 0.140 | 0.040 | 0.040 | <0.008 |
| Chemical compound mg/m$^3$ | Acetone | <0.017 | 0.170 | <0.017 | 0.240 |
| | *After the Implementation of the Deodorization System (Period II)* | | | | |
| | Ammonia | 0.127 | 0.075 | 0.236 | 0.104 |
| | $H_2S$ | <0.028 | <0.028 | <0.028 | <0.028 |
| | Ethyl mercaptan | <0.006 | <0.006 | <0.006 | <0.006 |
| | Butyl mercaptan | <0.008 | <0.008 | <0.008 | <0.008 |
| | Acetone | <0.017 | <0.017 | <0.017 | <0.017 |

In the calculation section of the pollution spread, meteorological data were taken on the basis of the "Catalogue of Meteorological Data, Guidelines for calculating the state of atmospheric air pollution", for the nearest station, Legnica:

anemometer height: 14 m
annual average temperature: 281.6 K
in accordance with Annex 4 to the Regulation of the Minister of the Environment of 26 January 2010, on reference values for certain substances in the air (Journal of Laws 2010 No. 16 item 87, as amended) [12].

According to the requirements of the reference methodology model, the average terrain roughness coefficient was determined for an area with a surface diameter equal to fifty times the highest emitter within the plant. The value of the aerodynamic coefficient of terrain roughness was calculated in accordance with the regulation on reference values for certain substances in the air [12], i.e., as a weighted average for the area concerned, calculated based on the value of the terrain roughness around the site concerned for each area type. In the area for which the calculations were carried out, there are no residential buildings $z_0$ (year) = 0.035 m.

Modelling was performed based on indicators calculated on the basis of measurements carried out at the plant (Table 4).

**Table 4.** Emissions of individual pollutants at each measuring point during period I.

| Compound Name | Ammonia | $H_2S$ | Ethyl Mercaptan | Buthyl Mercaptan | Acetone |
|---|---|---|---|---|---|
| **Emitter EP1** Green Waste Storage Yard | | | | | |
| Compound concentration, $\mu g/m^3$ | 210 | 28 | 16 | 14 | 17 |
| Air stream in tunnel, $m^3/s$ | 0.000138 | 0.00013 | 0.000138 | 0.000138 | 0.000138 |
| Wind tunnel area, $m^2$ | 0.00384 | 0.00384 | 0.00384 | 0.00384 | 0.00384 |
| WS, $\mu g/s/m^2$ | 7.55 | 1.01 | 0.58 | 0.50 | 0.61 |
| Area of emission source, $m^2$ | 180 | 180 | 180 | 180 | 180 |
| Ammonia emissions from the source, kg/h | 0.00489 | 0.00065 | 0.00037 | 0.00033 | 0.00040 |
| Emitter height, m | 4 | 4 | 4 | 4 | 4 |
| Gas temperature, K | 297.1 | 297.1 | 297.1 | 297.1 | 297.1 |
| **Emitter EP2** COMPOST Storage Yard | | | | | |
| Compound concentration, $\mu g/m^3$ | 85 | 28 | 55 | 40 | 170 |
| Air stream in tunnel, $m^3/s$ | 0.000138 | 0.00013 | 0.000138 | 0.000138 | 0.000138 |
| Wind tunnel area, $m^2$ | 0.00384 | 0.00384 | 0.00384 | 0.00384 | 0.00384 |
| WS, $\mu g/s/m^2$ | 3.05 | 1.01 | 1.98 | 1.44 | 6.11 |
| Area of emission source, $m^2$ | 1600 | 1600 | 1600 | 1600 | 1600 |
| Ammonia emissions from the source, kg/h | 0.01760 | 0.00580 | 0.01139 | 0.00828 | 0.03519 |
| Emitter height, m | 1.5 | 1.5 | 1.5 | 1.5 | 1.5 |
| Gas temperature, K | 293.1 | 293.1 | 293.1 | 293.1 | 293.1 |
| **Emitter EP3** Mixed Waste Storage Yard | | | | | |
| Compound concentration, $\mu g/m^3$ | 400 | 28 | 55 | 40 | 17 |
| Air stream in tunnel, $m^3/s$ | 0.000138 | 0.0001 | 0.000138 | 0.000138 | 0.000138 |
| Wind tunnel area, $m^2$ | 0.00384 | 0.00384 | 0.00384 | 0.00384 | 0.00384 |
| WS, $\mu g/s/m^2$ | 14.38 | 1.01 | 1.98 | 1.44 | 0.61 |
| Area of emission source, $m^2$ | 630 | 630 | 630 | 630 | 630 |
| Ammonia emissions from the source, kg/h | 0.03260 | 0.00228 | 0.00448 | 0.00326 | 0.00139 |
| Emitter height, m | 4 | 4 | 4 | 4 | 4 |
| Gas temperature, K | 298.1 | 298.1 | 298.1 | 298.1 | 298.1 |
| **Emitter E4** Sorting Facility | | | | | |
| Compound concentration, $\mu g/m^3$ | 110 | 28 | 6 | 8 | 240 |
| Air stream in tunnel, $m^3/s$ | 0.000138 | 0.00013 | 0.000138 | 0.000138 | 0.000138 |
| Wind tunnel area, $m^2$ | 0.00384 | 0.00384 | 0.00384 | 0.00384 | 0.00384 |
| WS, $\mu g/s/m^2$ | 3.95 | 1.01 | 0.22 | 0.29 | 8.63 |
| Area of emission source, $m^2$ | 12 | 12 | 12 | 12 | 12 |
| Ammonia emissions from the source, kg/h | 0.00017 | 0.00004 | 0.00001 | 0.00001 | 0.00037 |
| Emitter height, m | 15 | 15 | 15 | 15 | 15 |
| Gas temperature, K | 295.1 | 295.1 | 295.1 | 295.1 | 295.1 |

*2.2. Data Analysed*

The data on the efficiency of the plant and the energy consumed were obtained directly from the plant operator (Table 1), the MBP plant, and from documentation for granting the integrated permit.

The data on odor compound concentrations in specific periods of the plant's operation were obtained by means of environmental measurements and chromatographic tests [13]. The periods of operation were determined on the basis of data concerning plant operation before and after the implementation of the deodorization system.

The modelling included all the following substances: ammonia, hydrogen sulfide, ethyl mercaptan, butyl mercaptan, and acetone. The compounds included in the modelling are foul-smelling substances that are most common in the municipal waste and green waste treatment sectors.

### 2.3. Technological Parameters of the Plant

The core activity of the plant, located in the Lower Silesian region (Figure 1), is the mechanical and mechanical-biological processing of waste, including mixed municipal waste, and the separation from mixed municipal waste of fractions suitable in whole or in part for recycling, sorting, and containing raw material waste, including packaging waste, collected selectively, as well as the processing of selectively collected green waste and other biowaste.

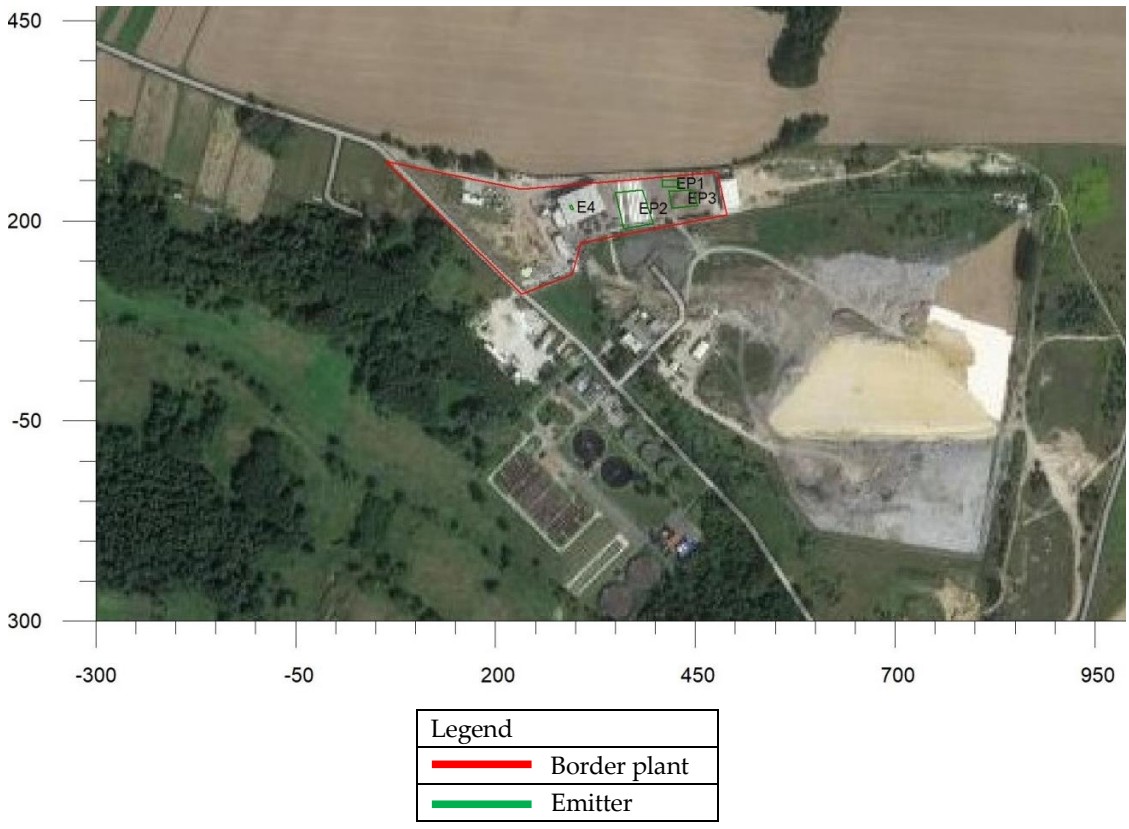

**Figure 1.** Location of the plant and emitters.

Various types and groups of waste can be mechanically treated. The plant has been designed and constructed to enable independent processing with the use of its individual units, e.g., only shredding, magnetic separation, or screening. The plant has the following main components:

−   Waste crusher (bag opener);
−   Sifter separating fractions 0–80 mm and above;
−   Sorting cabin with a conveyor system;
−   Magnetic separator;
−   Control room;
−   Sifter separating fractions 0–20 mm and above with a conveyor system and a loading system;

−   Magnetic separator.

Biological waste treatment relies on biological stabilization in the maturation area and composting in closed aeration reactors. In closed, ventilated reactors, the process of biological, aerobic waste treatment takes place, i.e., bio-stabilization, until the parameters resulting from the relevant provisions are reached. Biological waste treatment involves the integration of the following elements:

−   A sealed concrete slab with a system for collecting and draining leachate (industrial effluent)—a designated location for bio-stabilization reactors (surface area: 2000 $m^2$);
−   A specialized, self-propelled machine—an in-line loader for filling reactors;
−   Faceplates—stationary steel plates at the reactor inlet, where connections and fan mounting are located;
−   One aeration fan (1–4 kW) for each reactor;
−   A system of perforated pipes laid in the reactor, i.e., two aeration pipes that can be used to increase the humidity level of the processed waste, and one pipe for the removal of process air or condensate;
−   Temperature sensors, probes for monitoring process parameters in the reactors;
−   A central, microprocessor-based system that controls and monitors the biological waste treatment process;
−   Bio-reactor, a disposable bioreactor made of a special, three-layer, UV-resistant polyethylene film with very high tear- and perforation-resistance.

The above list of data is an integral element of determining the operation of the plant (Table 1). Actual plant performance corresponds to actual amount of waste (official SI unit is the megagram (symbol: Mg)) subjected to treatment processes (process D8, R3), registered by the plant management in 2015 and 2019 years (periods I and II respectively). Processes that have changed were selected for the analysis. Processes were defined according to Annex of the Act of 14 December 2012 on waste (Journal of Laws of 2013, item 21, as amended) [14]. Process D8 is a biological treatment that produces the final compounds or mixtures that are rendered harmless by any of the processes listed in D1–D12. Process R3 is the recycling or regeneration of organic substances that are not used as solvents (including composting and other biological transformation processes).

*2.4. Odor Compound Emissions*

Chromatographic tests were performed at the emission sites identified during previous local inspections (Table 2).

Samples for testing the concentrations of substances were taken using a kit consisting of suitable adsorbents through which air was drawn in by means of a stabilized flow aspirator (ASP-3 I) using the active carbon sorption method, which was prepared in accordance with PN-90/C-97554. The methodology for determining organic compounds adsorbed on active carbon is based on chromatographic analysis that uses columns filled with suitable separation material. At each point, one average sample was taken for analysis for the presence of the test substances. The sampling time totaled a minimum of 3 h. Air parameters were measured during sampling.

Organic substances were divided into groups according to the analytical methodology used for their determination:

−   Determination of hydrogen sulfide and mercaptans concentrations: gas samples were taken into Flex Foil bags, from which the gases were then extracted and introduced into a gas chromatograph with an FDP detector;
−   Determination of hydrogen sulfide concentrations: the results of measurements of sulphur compounds in the first series showed an absence of these substances in the air. Therefore, it was decided to use Draeger indicator tubes in the second series of tests;

－　Determination of ammonia concentrations in gases—air was passed through deionized water, which was then fed to the ion chromatograph.

The results of the substance concentration measurements during the test period are given below (Table 3).

Emissions were calculated for the concentration values in order to determine the proportion of individual emissions at each measurement point (Table 4, Table 5). The following tables show the calculated emissions together with the characteristics of the emitter.

**Table 5.** Emissions of individual pollutants at each measuring point during period II.

| | | | | | |
|---|---|---|---|---|---|
| **Emitter EP1**<br>**Green Waste Storage Yard** | | | | | |
| **Compound Name** | **Ammonia** | **H₂S** | **Ethyl Mercaptan** | **Butyl Mercaptan** | **Acetone** |
| Compound concentration, $\mu g/m^3$ | 127 | 28 | 6 | 8 | 17 |
| Air stream in tunnel, $m^3/s$ | 0.000138 | 0.00013 | 0.000138 | 0.000138 | 0.000138 |
| Wind tunnel area, $m^2$ | 0.00384 | 0.00384 | 0.00384 | 0.00384 | 0.00384 |
| WS, $\mu g/s/m^2$ | 4.56 | 1.01 | 0.22 | 0.29 | 0.61 |
| Area of emission source, $m^2$ | 180 | 180 | 180 | 180 | 180 |
| Ammonia emissions from the source, kg/h | 0.00296 | 0.00065 | 0.00014 | 0.00019 | 0.00040 |
| Emitter height, m | 4 | 4 | 4 | 4 | 4 |
| Gas temperature, K | 302.1 | 302.1 | 302.1 | 302.1 | 302.1 |
| **Emitter EP2**<br>**Compost Storage Yard** | | | | | |
| Compound concentration, $\mu g/m^3$ | 75 | 28 | 6 | 8 | 17 |
| Air stream in tunnel, $m^3/s$ | 0.000138 | 0.00013 | 0.000138 | 0.000138 | 0.000138 |
| Wind tunnel area, $m^2$ | 0.00384 | 0.00384 | 0.00384 | 0.00384 | 0.00384 |
| WS, $\mu g/s/m^2$ | 2.70 | 1.01 | 0.22 | 0.29 | 0.61 |
| Area of emission source, $m^2$ | 1600 | 1600 | 1600 | 1600 | 1600 |
| Ammonia emissions from the source, kg/h | 0.01553 | 0.00580 | 0.00124 | 0.00166 | 0.00352 |
| Emitter height, m | 1.5 | 1.5 | 1.5 | 1.5 | 1.5 |
| Gas temperature, K | 302.1 | 302.1 | 302.1 | 302.1 | 302.1 |
| **Emitter EP3**<br>**Mixed Waste Storage Yard** | | | | | |
| Compound concentration, $\mu g/m^3$ | 236 | 28 | 6 | 8 | 17 |
| Air stream in tunnel, $m^3/s$ | 0.000138 | 0.00013 | 0.000138 | 0.000138 | 0.000138 |
| Wind tunnel area, $m^2$ | 0.00384 | 0.00384 | 0.00384 | 0.00384 | 0.00384 |
| WS, $\mu g/s/m^2$ | 8.48 | 1.01 | 0.22 | 0.29 | 0.61 |
| Area of emission source, $m^2$ | 630 | 630 | 630 | 630 | 630 |
| Ammonia emissions from the source, kg/h | 0.01924 | 0.00228 | 0.00049 | 0.00065 | 0.00139 |
| Emitter height, m | 4 | 4 | 4 | 4 | 4 |
| Gas temperature, K | 301.1 | 301.1 | 301.1 | 301.1 | 301.1 |
| **Emitter E4**<br>**Sorting Facility** | | | | | |
| Compound concentration, $\mu g/m^3$ | 104 | 28 | 6 | 8 | 17 |
| Air stream in tunnel, $m^3/s$ | 0.000138 | 0.00013 | 0.000138 | 0.000138 | 0.000138 |
| Wind tunnel area, $m^2$ | 0.00384 | 0.00384 | 0.00384 | 0.00384 | 0.00384 |
| WS, $\mu g/s/m^2$ | 3.74 | 1.01 | 0.22 | 0.29 | 0.61 |
| Area of emission source, $m^2$ | 12 | 12 | 12 | 12 | 12 |
| Ammonia emissions from the source, kg/h | 0.00016 | 0.00004 | 0.00001 | 0.00001 | 0.00003 |
| Emitter height, m | 15 | 15 | 15 | 15 | 15 |
| Gas temperature, K | 305.1 | 305.1 | 305.1 | 305.1 | 305.1 |

Data constituting the calculation of odor compounds emissions were collected under real conditions, during sampling.

## 2.5. Deodorization System

The deodorization system consists of a sequence of techniques that lead to a reduction in odor compounds and substances in the form of an integrated system [15,16]. The deodorization system implemented within the plant consists of the following elements:

1. A system of "dry" spraying with an essential oil and acetate-based anti-odor agent, installed in the sorting facility. Within the facility's structure was suspended an odor-neutralizing steam generation system—DDG 500 diffuser, which is used to produce anhydrous mist, which neutralizes foul-smelling gas. The mist is used to neutralize noxious odors during the unloading, temporary storage, and mechanical processing of municipal waste.

2.  Preparation of waste for the bio-stabilization process using a biopreparation called OWS. Undersized waste is grafted with a special composition of active micro-organisms (OWS), which are used to accelerate the process of organic matter mineralization, reducing the weight and humidity of the processed waste, which primarily results in a significant minimization of the emission of foul-smelling compounds both in the mechanical and biological processes, as well as in the processing and management of the stabilized compost. The undersized waste installation includes a spraying system (FAM 500 B), a dosing device that maintains a constant temperature and solution concentration and continuously doses the exact amount of the solution of bio-active bacteria and water to the nozzles above the transport line.

3.  A biological filter, a device for post-process air purification, where a stream of waste gases from reactors (sleeves) passes through a bed of organic material, where it is biologically oxidized by micro-organisms into carbon dioxide, water, inorganic salts, and biomass. The biological filter is connected to an appropriate ventilation and air circulation system, in order to ensure even air distribution in the filling and sufficient waste gas dwell time in the bed.

4.  Self-propelled mobile spraying unit with an anti-odor agent. The entire plant area is sprayed with a spray nozzle mounted on an agricultural tractor with a special biological agent to ensure the hygienization and deodorization of the waste treatment site.

5.  Cleaning of the storage and handling yard with the use of a scouring and suction device, which cleans the surfaces where the biological waste treatment takes place (at a frequency of up to 560 h per year or as needed, depending on the state of pollution).

6.  The removal of small contaminants in the form of solid and liquid waste on the plant premises; in the case of waste (foul-smelling substances) gathering in hard-to-reach places, they are removed by means of an industrial vacuum cleaner with a separator to collect contaminants.

The system was implemented under operational conditions. The capital cost of the deodorization system is approximately 150,000.00 EUR.

The above-mentioned elements of the deodorization system are classified according to the standards laid down in Commission Implementing Decision (EU) 2018/1147 for waste treatment, and comply with BAT 12, 13, and 14 [17].

## 3. Results

Modeling was performed on the basis of the estimated emissions of individual pollutants at each measuring point during period I, II (Table 3; Table 4). The dispersion of air pollutants in the atmosphere is described using the plume model and the Pasquille equation. Based on this model and statistics on meteorological situations, the probability of exceeding certain pollutant levels during the year was determined [18,19]. Detailed calculations were made in accordance with Annex 4 to the Regulation of the Minister of the Environment of 26 January 2010, on reference values for certain substances in the air (Journal of Laws 2010 No. 16 item 87, as amended) [12]. The degree of emission reduction indicates a clear change in the value of individual chemical compounds before and after the implementation of the deodorization system in question, which is shown in the diagram below (Figure 2). The percentage change in the value of the process parameters was calculated on the basis of Table 1. The result is shown in the figure below (Figure 3).

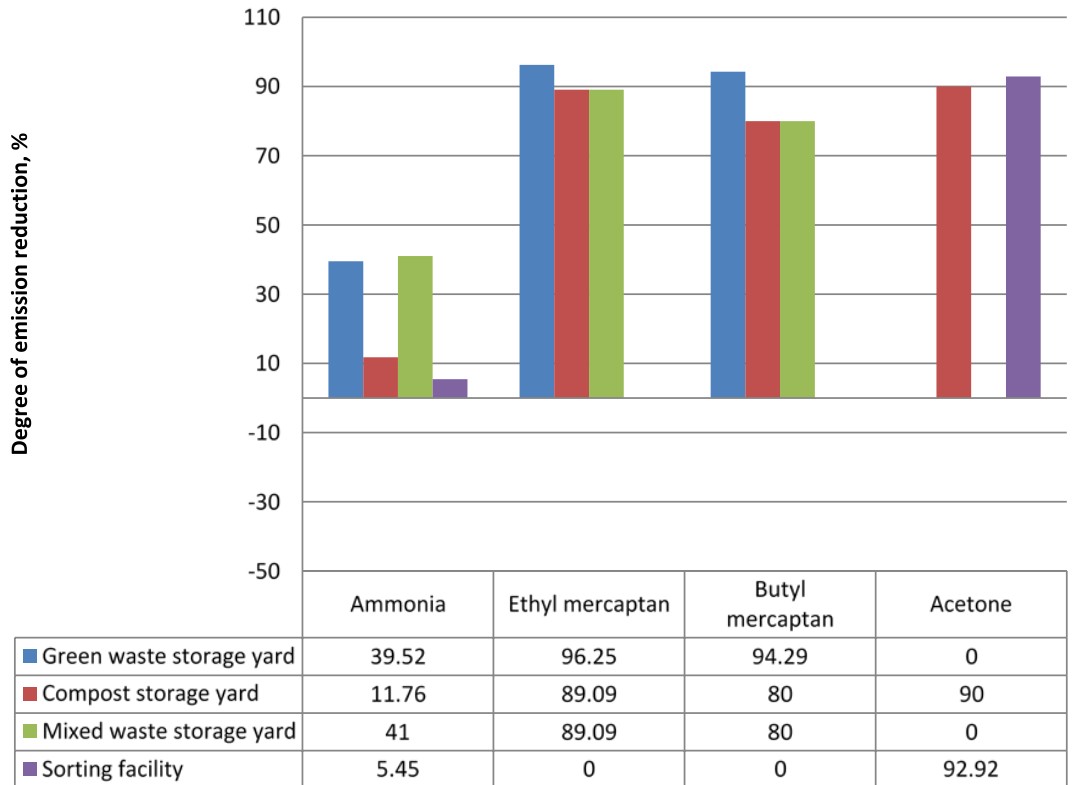

| | Ammonia | Ethyl mercaptan | Butyl mercaptan | Acetone |
|---|---|---|---|---|
| ■ Green waste storage yard | 39.52 | 96.25 | 94.29 | 0 |
| ■ Compost storage yard | 11.76 | 89.09 | 80 | 90 |
| ■ Mixed waste storage yard | 41 | 89.09 | 80 | 0 |
| ■ Sorting facility | 5.45 | 0 | 0 | 92.92 |

**Figure 2.** Degree of emission reduction following implementation of the deodorization system.

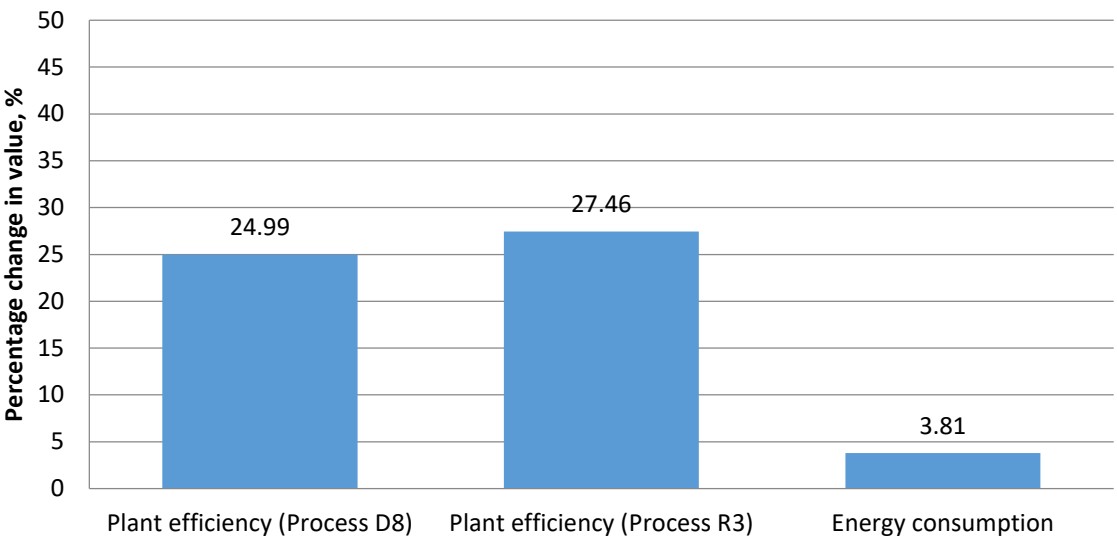

■ Change of the process parameter values after the implementation of the deodorization system

**Figure 3.** Change in the process parameter values after system implementation.

The dispersion of the emissions of individual pollutants in the atmosphere were made on the basis of a configured grid with a pitch of 20 m from each point in the X, Y coordinate system. Based on the calculated modeling, results were made in terms of isolines for each substance and are derived and shown in the figures ((Figures 4–6).

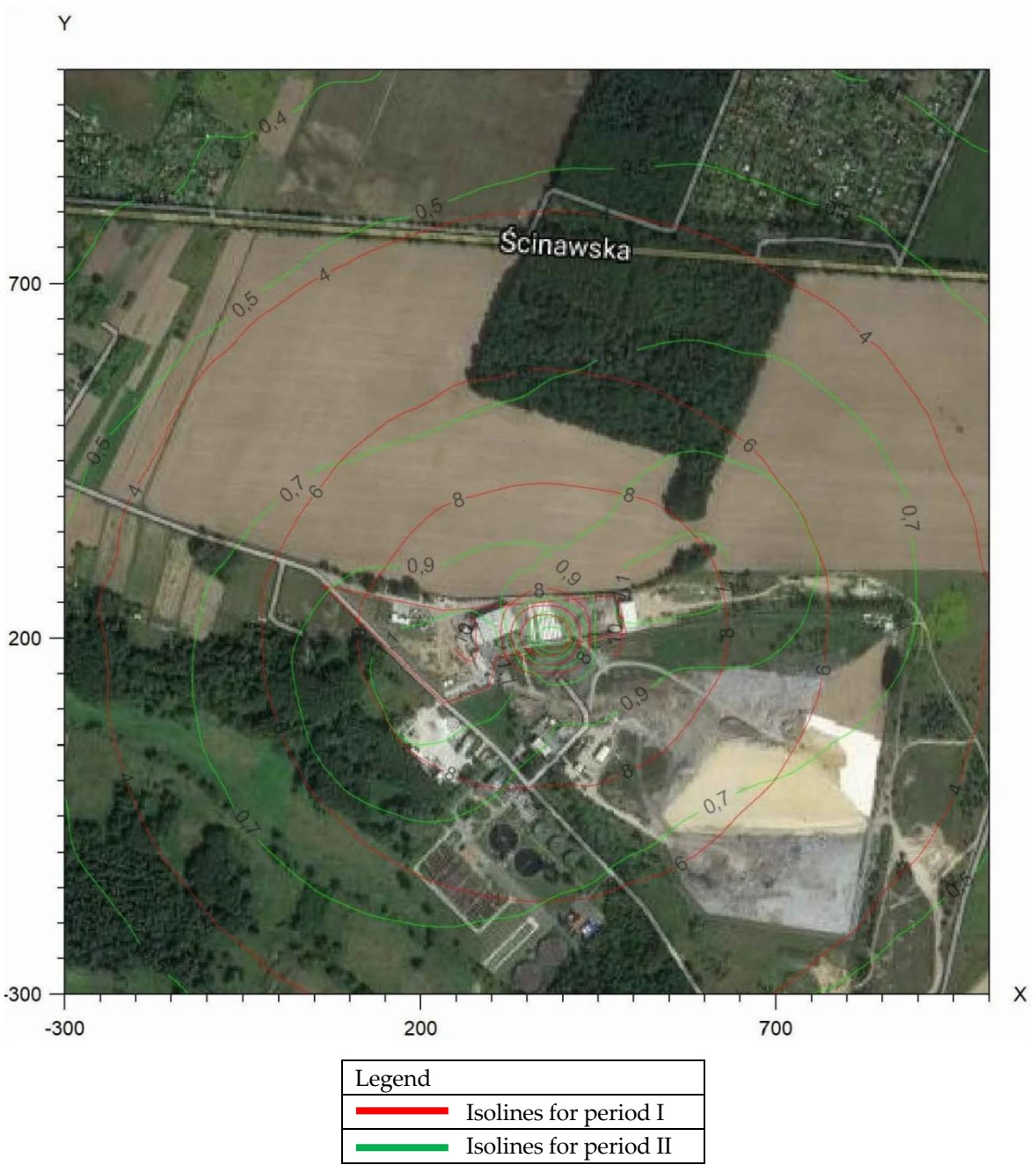

**Figure 4.** Isolines of the maximum acetone concentration µg/m$^3$ (in relevant periods).

The degree of emission reduction was calculated on the basis of Table 3.

In order to determine the total reduction of the odor compounds determined on the basis of the above measurements over the relevant periods of time, while taking into account the meteorological conditions, modelling of the airborne spread of pollutants was performed in accordance with the reference method [12].

The modelling results are shown in the diagrams below (Figures 4–6). The concentration of hydrogen sulfide has not changed, and therefore is not shown in the diagram.

Moreover, on the basis of the modelling performed, the predicted effect was calculated in order to precisely determine the distribution of the concentration of odor compounds in the environment, including the definition of the total reduction rate in odorous compounds (Table 6).

The calculations were based on the total value of mercaptans, which consists of butyl and ethyl mercaptan.

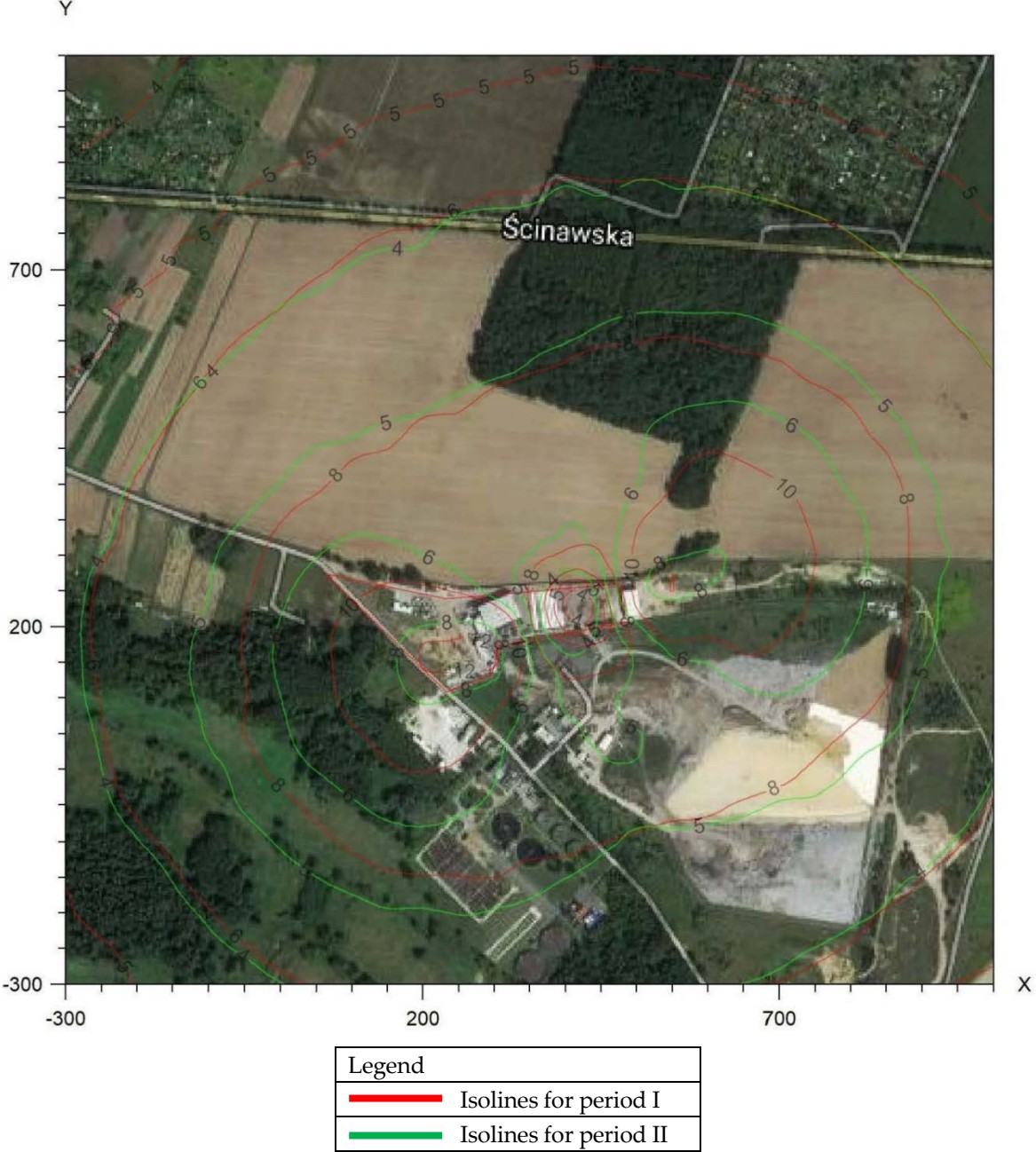

**Figure 5.** Isolines of the maximum ammonia concentration μg/m$^3$ (in relevant periods).

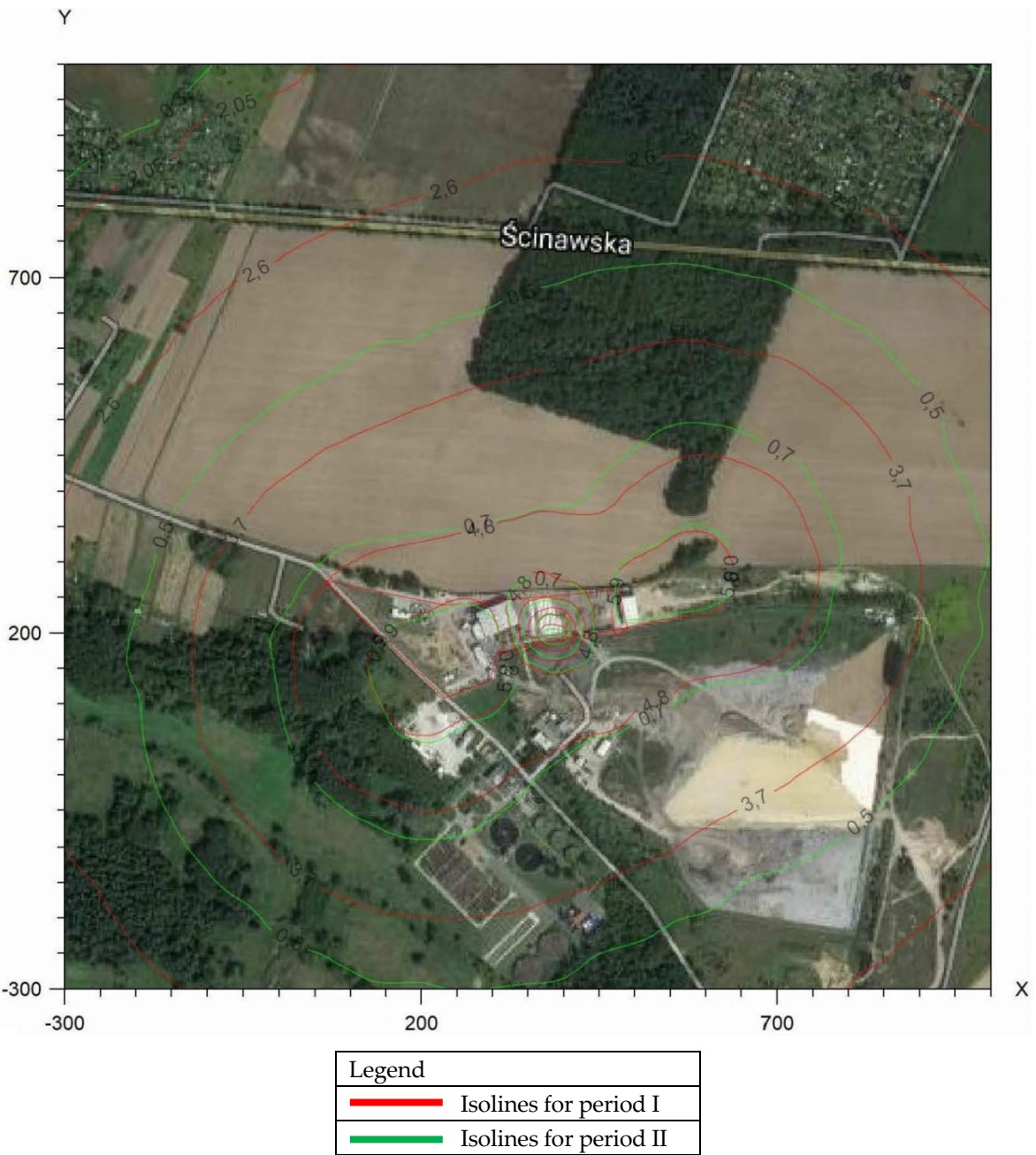

**Figure 6.** Isolines of the maximum mercaptans concentration µg/m$^3$ (in relevant periods).

**Table 6.** Summary of the total reduction in emissions.

| Name of Substance | Status for Period I, Emissions Volume Mg/year | Status for Period II, Emissions Volume Mg/year | Emission Reduction Mg/year | Total Emission Reduction % |
|---|---|---|---|---|
| Ammonia | 0.48410 | 0.33190 | 0.15220 | 31.44 |
| Mercaptans | 0.24642 | 0.03846 | 0.20796 | 84.39 |
| Acetone | 0.32720 | 0.04678 | 0.28042 | 85.70 |

## 4. Discussion and Conclusions

The issue of sustainable development of the surrounding space in the sphere of waste management industry requires consideration of many economic, environmental, technological, and social aspects. The result of the research work carried out as part of measurements of selected odorants such as

mercaptans, hydrogen sulfide, ammonia, and acetone, before implementation of the deodorization system and after implementation of the deodorization system, shows the relative potential impact on sustainable development factors in waste management.

"Can the deodorization system be used for other waste treatment facilities?" Designing and implementing systems that affect the technological system requires consideration of all variable parameters to determine the appropriate configuration of the technical solution. The issue was discussed in the scope of the configured deodorization system for the relevant technological parameters of the selected plant. The analytical methodology in the future can be a tool to for enabling the configuration and modernization of relevant technological systems in the waste management industry.

The changes shown as a result of the described implemented system lead to the determination of the possible environmental, economic, and technological effects, as well as to the behavior assessment of the interdependent variables, and allow one to conclude that the implemented solutions are effective environmentally and technologically. Total emission reduction helps to estimate the potential environmental effect that depends on social and economic aspects. The result of the analysis carried out may constitute a reference value in estimating possible changes of technological systems and implemented emission reduction solutions for planned mechanical-biological waste treatment facility. Deeper involvement of the scientific community in this issue in this field of research could be beneficial to both stakeholders of companies and citizens.

The assumed analytical methodology can be the basis for further research in determining the main aspects of sustainable development for the waste management industry in case of emission reduction.

The above analyses indicate that, due to implementing a deodorization system for the municipal waste processing plant under analysis, the following effects were obtained:

Total emissions were reduced as follows:

(a) 84.39% for mercaptans;
(b) 31.44% for ammonia;
(c) 85.70% for acetone.

The concentration of hydrogen sulfide has not changed.
The efficiency of the plant was increased by

(a) 24.99% (+14,920.64 Mg/year) in the D8 process;
(b) 27.46% (+1045.1 Mg/year) in the R3 process.

The demand for electricity increased by 3.81% (+18,618 kWh).
The economic effect is as follows:

(a) The annual operating cost based on electricity consumption will be approximately 2792.7 EUR/year (assuming a market price for electricity of 0.15 EUR/kWh);
(b) The capital cost of the deodorization system is approximately 150,000.00 EUR;
(c) Profit associated with increased efficiency will be approximately 282,593.60 EUR/year (assuming 20% of the market price from waste processing of 17.7 EUR/Mg).

In conclusion, introducing a deodorization system on the plant's premises contributed to the optimization of technological processes and, as a consequence, to the achievement of tangible ecological and economic effects in terms of increasing the plant's operational efficiency (despite the increased demand for electricity). The implementation of the deodorization system had a positive impact on the parameters determining the functioning of the plant—the mechanical-biological waste treatment facility.

**Funding:** This research received no external funding.

**Acknowledgments:** 

**Conflicts of Interest:** The authors declare no conflict of interest.

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
