# Peer review of "Impact of Implementing a Deodorization System on the Functioning of a Waste Management Plant"

_sustainability, doi:10.3390/su12155983_

Round 1

Reviewer 1 Report

The reviewed article is factually correct. Refers to current issues. The author presented a case study. The purpose of this work is the evaluation of the impact of the deodorisation system on the parameters determining the functioning of the plant - a mechanical-biological waste treatment facility in order to determine sustainability factors and their relative changes. The goal has been achieved. The analysis is generally correct. Correct conclusions. These are the positives of the study.
Suggestions for improvement:
1) If possible, I suggest to give the real name of the company (waste management plant).
2) I suggest completing the summary with the conclusion of the analysis.
3) I suggest limiting the number of keywords to 5 words.
4) I suggest using more of the literature reviewed / analyzed in the introduction.
5) Line 60 and 61 - no punctuation.
6) In the text before the table there should be a reference to the table number. Unfortunately, these references are missing throughout the article.
7) I suggest not finishing the subsection (e.g. 2.3, 2.4) with the table and the text.
8) As in the case of tables, drawings should be cited in the text before them.
9) Lines 249-256. Bullet edit error.
I don't feel qualified to judge about the English language and style.
After corrections have been made, the article can be published as a case study.

Author Response

Hello,

Dear reviewer, thank you for the valuable comments that I have read. I added additions to the article according to all comments. Regarding 1 remark - the name of the plant is private information. The changes to the article also take into account the comments of the second reviewer.

Best regards,

Łukasz Szałata

Reviewer 2 Report

The paper discusses the impact of a deodorization system on a waste management plant before and after the implementation of the system. The results could be interesting to readers working in the area. However, some of the contents lack clarification and in-depth discussion. I would suggest a major revision to improve the quality. I have the following comments regarding the details.

  1. In the introduction, it is missing if similar case studies have been conducted for other systems and why the case study in this paper is important for readers.
  2. In Section 2.3, it would be great to include a 2D plant figure to describe the location of each functioning units and pointing out the sampling locations to help readers better understand general information about the plant.
  3. In Table 1, the actual plant performance has a unit of Mg/year. What is the actual plant performance is defined? Is the unit correct? It needs some explanations. Also, it is quite confusing to suddenly have the Process D3 and R3 without any explanation. Also in Table 6, the emission volume unit is taken as Mg.
  4. Table 2 seems unnecessary as Table 3 also includes the location information.
  5. Line 130, “The 130 sampling time totalled a minimum of 3 hours.” Roughly how much volume of air were sampled?
  6. Line 204, Please briefly describe how the modeling was conducted. The readers could go to the reference for more details but it would be better to have basic description here.
  7. For Figure 3-6, how were the concentration profiles derived and how were the profiles used for calculating the emissions?
  8. Hydrogen sulfide (please correct sulphide to sulfide) is not accurately determined in the first place (only a rough range) Therefore, it is not necessary to include it in Table 6 and Figure 1. And how was the H2S concentration in Figure 5 determined? You do not have the accurate concentration of H2S in those four sampling locations, right?
  9. If possible, it would be great to include the capital cost for the deodorization system. The electricity could be considered as operation cost, it would be very helpful to have the capital cost to have a general idea of cost for the system.

Author Response

Hello,

Dear reviewer, thank you for the valuable comments that I have read. I added changes to the article according to all comments. Below I am sending answers to questions that were not included in the revised version of the article:
- Regarding 5 question - Sampled volume of air can be calculated from air stream in tunnel and sampling time (in this case it is about 90 m3). In the sampling methodology used, the air flows through a layer of appropriate separating material.
- Regading one of question from 8 - The concentration of hydrogen sulfide was too low - beyond the limit of quantification.

The changes to the article also take into account the comments of the second reviewer.

The corrected article is uploaded.

Best regards,

Łukasz Szałata

Round 2

Reviewer 2 Report

I see most of my concerns addressed for the current version and think the quality is good for publication. Please include a changed tracked version for reviewers in the future. It will make things easier. That could be easily done by comparing your revised version with your original version in Word.